# Robust Attitude and Heading Estimation under Dynamic Motion and Magnetic Disturbance

**DOI:** 10.3390/mi14051070

**Published:** 2023-05-18

**Authors:** Fan Bo, Jia Li, Weibing Wang, Kaiyue Zhou

**Affiliations:** 1Institute of Microelectronics of the Chinese Academy of Sciences, Beijing 100029, China; bofan@ime.ac.cn (F.B.); wangweibing@ime.ac.cn (W.W.); zhoukaiyue@ime.ac.cn (K.Z.); 2School of Microelectronics, University of Chinese Academy of Sciences, Beijing 100049, China

**Keywords:** MEMS IMU, attitude estimation, AHRS, complementary filter, magnetic disturbance, deep learning

## Abstract

Robust and accurate attitude and heading estimation using Micro-Electromechanical System (MEMS) Inertial Measurement Units (IMU) is the most crucial technique that determines the accuracy of various downstream applications, especially pedestrian dead reckoning (PDR), human motion tracking, and Micro Aerial Vehicles (MAVs). However, the accuracy of the Attitude and Heading Reference System (AHRS) is often compromised by the noisy nature of low-cost MEMS-IMUs, dynamic motion-induced large external acceleration, and ubiquitous magnetic disturbance. To address these challenges, we propose a novel data-driven IMU calibration model that employs Temporal Convolutional Networks (TCNs) to model random errors and disturbance terms, providing denoised sensor data. For sensor fusion, we use an open-loop and decoupled version of the Extended Complementary Filter (ECF) to provide accurate and robust attitude estimation. Our proposed method is systematically evaluated using three public datasets, TUM VI, EuRoC MAV, and OxIOD, with different IMU devices, hardware platforms, motion modes, and environmental conditions; and it outperforms the advanced baseline data-driven methods and complementary filter on two metrics, namely absolute attitude error and absolute yaw error, by more than 23.4% and 23.9%. The generalization experiment results demonstrate the robustness of our model on different devices and using patterns.

## 1. Introduction

With the widespread use of smartphones, wearable devices, wireless earphones, head-mounted VR devices, Micro Aerial Vehicles (MAVs), and vehicle Advanced Driving Assistance Systems, the small and low-cost inertial sensors based on Micro-Electromechanical System (MEMS) technology, also known as MEMS Inertial Measurement Units (MEMS-IMUs), are gradually playing an irreplaceable role in people’s lives. According to the latest research report, the consumer-grade IMU market is expected to grow to 838 million US dollars by 2026, with a compounded annual growth rate of 5% [1]. Using MEMS-IMU to track motion is crucial in various applications, such as pedestrian dead reckoning (PDR) [2,3], human activity recognition [4,5], health monitoring, rehabilitation robotics, and autonomous vehicles, which benefit fields such as the Internet of Things, smart cities, human–machine interaction, and medical health [6]. Any improvement in this technology will profoundly impact and extend a series of unprecedented systems and applications.

Due to limitations in the designing and manufacturing of MEMS sensors, the accuracy of MEMS-IMU has been regarded as challenging in achieving inertial navigation [7]. However, using MEMS-IMUs to obtain the attitude of the carrier has been an important application accompanying its development and mass adoption since [8]. The traditional method to estimate attitude is to use sensor fusion algorithms, such as the Attitude and Heading Reference System (AHRS), which uses motion data from 3-axis accelerometers, gyroscopes, and magnetometers, to calculate 3D attitudes [9]. The principle is to use the gravitational acceleration measured using the accelerometer during the stationary phase and the geomagnetic fields measured using the magnetometer as reference sources to correct for the error induced by the integration of the raw gyroscope measurement [10].

However, in practical scenarios, the accelerometer data represent the resultant vector of the carrier’s acceleration, which includes not only gravitational but also external acceleration generated by motion and vibrations caused by collisions. These interfering factors are ubiquitous during movement and are difficult to eliminate through modeling or other means due to their random nature [11]. Similarly, the actual data obtained using the magnetometer represent the resultant vector of the magnetic field at the carrier, which includes not only the geomagnetic field but also magnetic disturbances caused by ferromagnetic materials in local building structures [12]. These interfering factors vary with the region of use in both magnitude and direction, hindering the accurate measurement of the geomagnetic field. The aforementioned disturbances jointly damage the accurate and reliable estimation of attitude and heading. Therefore, exploring an AHRS algorithm that can mitigate the influence of large external acceleration and magnetic disturbance has become an emerging research direction [13,14,15,16].

### 1.1. Related Work

Traditional sensor fusion attitude estimators are mainly based on Kalman filtering (KF) [17] and a Vector Observation Algorithm [18]. However, KF and other optimization methods demand a high sensor sampling rate to achieve real-time prediction, and also require a large computational cost [19]. Researchers have developed sensor fusion algorithms with Complementary Filters (CF) [20] or Complementary Kalman Filter [21] structures to improve computational efficiency and have them applied to the AHRS. The adopted Mahony algorithm [20] is a typical CF algorithm in this field and is widely used in low-cost applications. Since the performance of CF algorithms depends largely on the selection of their gain parameters, a series of adaptive gain optimization methods have been proposed to improve the robustness of the AHRS based on complementary filtering [22]. Madgwick et al. proposed an AHRS algorithm based on the gradient descent algorithm (GDA) [19], which relies on the gradient calculated by the measured acceleration vector and the gravity vector pointing to the center of the earth to correct the accumulative error of the gyroscope. Kok et al. [23] fastened the convergence of the GDA through a single gradient descent step with fixed step length and further reduced the computational complexity by 36%. The advanced complementary filter MUSE [24] is similar to the Madgwick filter, but only uses the gravity vector as the attitude update source when the device is completely stationary. Instead, it mainly uses the geomagnetic north vector as the basis for gradient calculation, which eliminates the problem of erroneous updates caused by large external accelerations, as the acceleration vector is mainly composed of gravity when the device is stationary.

In 2020, Madgwick improved the original GDA algorithm and proposed an AHRS algorithm called the Extended Complementary Filter (ECF) [25], which is based on the Mahony’s original CF algorithm. The ECF algorithm uses both the gravity vector and geomagnetic north vector for attitude error correction. When the algorithm detects dynamic motion or magnetic disturbance, it disables the corresponding correction information and only utilizes the rest components for attitude updates. Fan et al. considered both magnetic disturbance and dynamic motion issues, first implementing a decoupled magnetic field complementary filter tilt estimation algorithm [26], and then proposing an adaptive anti-disturbance strategy based on a finite state machine [27]. Yuan et al. [28] modeled the uncertainty of external acceleration and magnetic disturbance and took them as the covariance matrix in an extended Kalman filter (EKF)-based attitude estimator. These algorithms’ anti-disturbance ability was validated in indoor human motion tracking experiments using well calibrated commercial modules, such as the MTw IMU sensor [29]. However, in diverse and low-cost MEMS inertial sensor devices, data quality is difficult to match that of commercial inertial modules, and the adaptively obtained parameters are difficult to transfer between different devices and usage scenarios, making such algorithms difficult to apply on a large scale. The VQF [30] decoupled the heading error from the inclination error in attitude state representation to reject magnetic disturbances and provided precise inclination correction using specific force measurements. The results on the publicly available dataset [31] showed greatly improved accuracy and robustness compared to traditional filtering methods.

With the development of deep learning and its advances in the field of robotics, more and more researchers are using neural networks to process inertial sensor data in the field of navigation, demonstrating strong fitting capabilities [32]. OriNet [33] and Brossard et al. [34] have started using deep networks to regress attitude from IMU measurements. OriNet uses a Long Short-Term Memory (LSTM) neural network to propagate states. It corrects gyroscope biases using genetic algorithms and mitigates sensor noise through additive Gaussian noise during training. Brossard et al. estimated attitude through gyroscope integration but correct angular velocity errors using CNNs to filter out unwanted noise and bias before integration. Both methods primarily focus on using deep networks to filter gyroscope data and estimating compensation factors to reduce bias and noise. Different from aforementioned methods, using an end-to-end framework to map attitude directly from raw IMU data has developed a filter that outperforms traditional filters in robustness, as shown in [35,36]. Extensive experiments on six publicly available datasets proved the ability of RIANN [35] to generalize over different application scenarios. IDOL [37] utilizes an LSTM to estimate both the attitude and covariance using nine-axis IMU data. At the same time, it uses the EKF to fuse gyroscope data and the neural network output in a short period of time to estimate the three-axis attitude.

Our work differs from previous works in the following two aspects. The first is that we model IMU calibration errors, especially dynamic motion and magnetic disturbance, using deep neural networks. The second is that we use an open-loop and decoupled version of the ECF framework to adaptively fuse denoised sensor data to provide accurate attitude estimation under different usage scenarios.

### 1.2. Our Contribution

In this paper, we propose a robust AHRS to adapt to different scenarios, especially under dynamic motion and magnetic disturbance. In summary, we have four key contributions:A detailed modeling of low-cost IMU calibration errors for data-driven error compensation including the gyroscope, accelerometer, and magnetometer.A series of Temporal Convolutional Network (TCN)-based random error estimation models that take raw data as the input and regress the error-compensated sensor data, including the denoised angular velocity, gravitational acceleration, and geomagnetic north field.An open-loop and error-decoupled version of the ECF framework for efficient and robust attitude estimation using a six-axis or nine-axis IMU.Extensive and qualitative experiments on three public datasets, covering different sensor platforms, motion status, and magnetic field environment. Additionally, we prove that our model outperforms the existing data-driven methods in both accuracy and robustness and is compact enough for implementation on resource-constrained devices.

The rest of this paper is organized as follows. Section 2 introduces the baseline method of this paper and defines the problems by rewriting the formulation of the ECF framework. Section 3 describes the details of the used calibration model, neural network architecture, overall optimization techniques, and the loss function. Section 5 presents the results of the experimental evaluations. Finally, the conclusion is drawn in Section 6.

## 2. Extended Complementary Filter Framework

We propose to use the ECF algorithm [25] as the sensor fusion framework. The main formulations of the ECF algorithm are introduced below, and we use a direction cosine matrix (DCM) to express attitude. When the 3D attitude and angular velocity of a rigid platform are known, the increment change in attitude can be expressed as
(1)$${\dot{C}}_{b}^{n}=C_{b}^{n}\Omega_{nb}^{b}$$
where $C_{b}^{n}\in \mathrm{SO}(3)$ is the DCM that maps the body coordinate frame (b-frame) to the North–East–Down navigation coordinate frame (n-frame), which is a member of the 3D rotation group (SO(3)), and $\Omega_{nb}^{b}$ is the anti-symmetric matrix of the angular velocity $\omega_{nb}^{b}={\left[\begin{array}{ccc} \omega_{x} & \omega_{y} & \omega_{z} \end{array}\right]}^{T}$, which is
(2)$$\Omega =[\omega \times ]=\left[\begin{array}{ccc} 0 & -\omega_{z} & \omega_{y}\\ \omega_{z} & 0 & -\omega_{x}\\ -\omega_{y} & \omega_{x} & 0 \end{array}\right]$$

Hence, the value for attitude at timestamp *k* can be calculated by integrating the increment change in the attitude ${\dot{C}}_{b}^{n}$, where exp(·) is the SO(3) exponential map.
(3)$$C_{b(k)}^{n}=C_{b(k-1)}^{n}\;\exp(\Omega_{nb}^{b}\Delta t)$$

Because the error of the gyroscope continues to accumulate over time through integration, the error term e is used for correction, and the global parameter K is the gain used to scale the error correction term. Then, Equation (1) for attitude update can be changed to
(4)$${\dot{C}}_{b}^{n}=C_{b}^{n}\left[\left(\omega +Ke\right)\times \right]$$

The error correction term e includes two parts: the accelerometer correction term $e_{a}$ and the magnetometer correction term $e_{m}$.
(5)$$e_{a}=\frac{a^{b}}{\left\|a_{b}\right\|}\times \left(C_{n(k-1)}^{b}\cdot \left[\begin{array}{c} 0\\ 0\\ 1 \end{array}\right]\right)$$
where ab=[axayaz]T are the measured acceleration in the b-frame, $C_{n(k-1)}^{b}$ is attitude estimated in last timestamp, and vector ${\left[\begin{array}{ccc} 0 & 0 & 1 \end{array}\right]}^{T}$ is the true gravity vector in the n-frame. The error term $e_{a}$ represents the difference between the measured gravitational acceleration in the zero-velocity range and the true gravity vector transformed in the b-frame. Therefore, the gravity vector can be used as a correction source for the attitude through the accelerometer.
(6)em=ab×mb‖ab×mb‖×(Cn(k−1)b·[010])
where mb=[mxmymz]T are the measured magnetic field in the b-frame, and vector ${\left[\begin{array}{ccc} 0 & 1 & 0 \end{array}\right]}^{T}$ is the unit vector pointing east in the n-frame, which represents the cross product of gravity vector and geomagnetic north vector to eliminate the influence of magnetic declination. Therefore, the vector pointing east can be used as a correction source for the heading angle through both the accelerometer and magnetometer. e can be described by a set of rules, as follows:(7)e={ea+emif ‖(‖ab‖−‖g‖)‖<amin  and  mmin<‖mb‖<mmaxeaelse if ‖(‖ab‖−‖g‖)‖<amin[000]Telse
where $a_{\min}$ is a small constant, g represents the gravity vector, and $\left[m_{\min},m_{\max}\right]$ represents the normal fluctuation range of the geomagnetic field. Equation (7) indicates that when the accelerometer detects large external acceleration or the magnetometer detects magnetic disturbances, the corresponding part is removed from the total error term to prevent erroneous correction. The gain *K* can be modeled as
(8)$$K=\{\begin{array}{ll} K_{norm} & \mathrm{if}\;t>t_{init}\\ K_{norm}+\frac{t_{init}-t}{t_{init}}\left(K_{init}-K_{norm}\right) & \mathrm{else} \end{array}$$
where $K_{init}$ is a large value for fast convergence to the correct attitude at the initialization period, and $K_{norm}$ is a smaller gain factor for normal operation of the model. By adjusting the parameter *K*, the algorithm can achieve the best performance, then realize the real-time solution to the attitude of the carrier.

However, simply removing the accelerometer and magnetometer calibration sources cannot alleviate the problem of gyroscope error accumulation. Using this approach for a prolonged period of time under dynamic motion and magnetic disturbance can still cause the attitude result to diverge. Additionally, the algorithm requires a relatively long static time for the attitude to converge, and short static or low-speed situations cannot reduce attitude error. Therefore, when deploying the algorithm in practical applications, corresponding improvements need to be made in this regard. Subsequent experiments have shown that the ECF algorithm is suitable for hardware systems with low gyroscope errors and high sampling rates. It is difficult to achieve good attitude prediction results for systems with low sampling rates and long periods of high dynamics and magnetic disturbance.

## 3. Denoising Neural Network for Robust AHRS

### 3.1. Low-Cost Inertial Measurement Unit (IMU) Calibration Model

The calibration model of low-cost IMU suffers from errors, which are either systematic errors or random errors. Systematic errors, such as a scale factor error, axis misalignment, and static bias, can be modeled mathematically and can be mitigated through calibration, while random errors are usually modeled as random processes within the inertial navigation solution scheme. [38] We introduce the IMU calibration model commonly used in developing the AHRS as follows [8]. The following terms are all in the b-frame. Gyroscope readout $y_{Gyr,k}$ at timestamp *k* can be modeled as
(9)$$y_{Gyr,k}=K_{Gyr}\omega_{k}+b_{Gyr,k}+v_{Gyr,k}$$
where $K_{Gyr}$ is the gyroscope scale factor matrix; $\omega_{k}$ is the true angular velocity in the b-frame; $b_{Gyr,k}$ is bias, which includes static bias $b_{Gyr,0}$ and time-varying random bias; and $v_{Gyr,k}$ is a white noise term. Accelerometer readout $y_{Acc,k}$ can be modeled as
(10)$$y_{Acc,k}=K_{Acc}\left[a_{k}+g\right]+b_{Acc,k}+v_{Acc,k}$$
where $K_{Acc}$ is the accelerometer scale factor matrix, $a_{k}$ is the external non-gravitational acceleration, g is gravitational acceleration, and $v_{Acc,k}$ is a white noise term. Magnetometer readout $y_{Mag,k}$ can be modeled as
(11)$$y_{Mag,k}=K_{Mag}m_{k}+d_{k}+v_{Mag,k}$$
where $K_{Mag}$ is the magnetometer scale factor matrix, $m_{k}$ is the true geomagnetic field, $d_{k}$ is magnetic distortion, and $v_{Mag,k}$ is a white noise term.

In practice, the aforementioned systematic error terms can be calibrated without external equipment during operation. Typically, the ellipsoid fitting method can be used in combination with the geomagnetic north vector and gravity vector to calibrate the magnetometer and accelerometer [39,40,41]. The gyroscope, on the other hand, needs to be fitted with the accelerometer using the hexahedron method [42] to obtain the gyroscope systematic errors during flipping, while the random errors can be calibrated by analyzing the zero bias changes during long-term static testing using the Allan variance [43]. Compared to the methods mentioned earlier, our paper presents a novel approach to correcting errors in IMUs using denoising neural networks. This data-driven method can address both systematic and random errors in IMUs. The concept is to utilize neural networks with trainable parameters to model error terms, enabling the mapping of raw sensor data to denoised physical quantities.

Leveraging the aforementioned common calibration models, we transform Equations (9)–(11) and obtain the denoised angular velocity ω^Gyr as
(12)ω^Gyr=R^GyryGyr,k−ω˜k
where R^Gyr∈ℝ3×3 is the transpose of matrix KGyr, which includes scale factor error and misalignment error of gyroscope, and error term ω˜k∈ℝ3 is mainly comprised of static and random bias bGyr,k. Similarly, the denoised gravitational acceleration g^k can be computed as
(13)g^k=R^AccyAcc,k−a˜k
where R^Acc∈ℝ3×3 is the transpose of matrix KAcc, and error term a˜k∈ℝ3 is mainly comprised of external acceleration ak. The denoised geomagnetic field b^k can be computed as
(14)b^k=R^MagyMag,k−m˜k
where R^Mag∈ℝ3×3 is the transpose of matrix KMag, and error term m˜k∈ℝ3×3 is mainly comprised of magnetic distortion.

### 3.2. Learning-Based Sensor Error Compensation

Now we seek to estimate error terms in IMU calibration models using data-driven methods. We define the neural network to estimate gyroscope error term ${\tilde{\omega }}_{Gyr}$ as
(15)$${\tilde{\omega }}_{Gyr}=f_{TCN}(y_{IMU,k-N},\ldots ,y_{IMU,k})$$
where neural network $f_{TCN}$ is similar to the network structure proposed by Brossard [34]. It is mainly composed of a five-layer 1D-CNN. The time series is processed by inputting N data frames before k time, and the sequence is serialized by padding zeros at the beginning of the sequence. Different expansion values are set in each layer of the CNN so that the receptive field of the highest layer can cover the entire sequence, and the entire network can realize the time window of N = 448. The CNN’s convolution kernel parameters gradually increase with the number of layers to achieve high-dimensional feature extraction, setting the Batch Normalization layer and Dropout layer after each layer to alleviate overfitting and using the GELU activation function to achieve nonlinear output. The structure and parameters of the network are shown in Figure 1.

As Equation (15) stated, the input data for the six-axis IMU are yIMU,k=(yGyr,k,yAcc,k)∈ℝ6, and for the nine-axis IMU are (yGyr,k,yAcc,k,yMag,k)∈ℝ9. It is important to include accelerometer and magnetometer data since the change in acceleration and magnetic field can also contain information for an incremental change in attitude. When the carrier is moving slowly and has no magnetic disturbance, the acceleration and magnetic field in the b-frame can be stated as
(16)akb=Cn(k)b(Δv/Δt+gn)
(17)mkb=Cn(k)b(Δd+bn)

Additionally, the change in acceleration and magnetic field can be computed as
(18)ak+1b−akb=Cn(k)bgn−Cn(k−1)bgn=(exp(−ΩnbbΔt)−I3)Cb(k−1)nTgn
(19)mk+1b−mkb=(exp(−ΩnbbΔt)−I3)Cn(k−1)bTbn
where gn is the gravity vector in the n-frame, and bn is the geomagnetic north vector in the n-frame. By providing multi-source data, the neural network can fuse information in different domains to improve the ability of the dynamic error compensation for angular velocity.

Similarly, we define the neural network to estimate error term ${\tilde{a}}_{k}$ as
(20)$${\tilde{a}}_{k}=f_{TCN}\left(y_{Acc,k-N},\ldots ,y_{Acc,k}\right)$$

The neural network to estimate error term ${\tilde{m}}_{k}$ as
(21)$${\tilde{m}}_{k}=f_{TCN}\left(y_{Mag,k-N},\ldots ,y_{Mag,k}\right)$$

Both neural networks utilize a TCN as the feature extractor.

Since the proposed calibration models are linear operations, in addition to the parameters of the neural network, other parameters of the calibration model, such as ${\hat{R}}_{Gyr}$, ${\hat{R}}_{Acc}$, and ${\hat{R}}_{Mag}$, can also be optimized through backpropagation. During initialization, the neural network outputs $0_{3}$, and ${\hat{R}}_{Gyr}$, ${\hat{R}}_{Acc}$, and ${\hat{R}}_{Mag}$ are set as $I_{3}$; error model outputs are, respectively, ${\hat{\omega }}_{Gyr}\approx y_{Gyr,k}$, ${\hat{g}}_{k}\approx y_{Acc,k}$, and ${\hat{b}}_{k}\approx y_{Mag,k}$, in order to speed up the training. After the training is completed, the obtained parameters ${\hat{R}}_{Gyr}$, ${\hat{R}}_{Acc}$, and ${\hat{R}}_{Mag}$ do not change with time, and they represent the scale factor and axis alignment error of the sensor.

Compared with the traditional calibration method, the parameters obtained through data driving can better reflect the real in-run error of the sensor. A follow-up will have experiments verify the accuracy of the calibration method.

### 3.3. Optimization for Open-Loop ECF Model

After acquiring denoised sensor data using the IMU calibration model, we utilize the ECF framework to fuse sensor data. The corrected angular velocity ω^k, which is used for attitude update, can be modeled as
(22)ω^k=ω^Gyr+γeAcc+ηeMag
where ω^Gyr is calculated using Equation (12), and γ∈ℝ3 and η∈ℝ3 are gain parameters. Error term eAcc can be calculated as
(23)eAcc=g^k‖g^k‖×(C˜n(k−1)b·[001])
where g^k is calculated using Equation (13), and eMag can be calculated as
(24)eMag=g′×b^k‖g′×b^k‖×(C˜n(k−1)b·[010])
where b^k is calculated using Equation (14).

According to the calculation of error correction terms from the accelerometer and magnetometer, the attitude value from the previous timestep must be obtained to achieve the transformation of the true reference vector. However, the serial closed-loop calculation will greatly increase the training time of the neural network, and the non-direct correspondence also increases the difficulty for training convergence. The Transformer [44] proposes to use masking to process the label sequence of the decoder input to achieve parallel calculation and accelerate the training of the decoder. However, due to the short time interval between adjacent outputs of the proposed model and the relatively simple dependency relationship, the multiplication of large masks undoubtedly increases the computational burden of the model.

We have taken the idea of the Transformer and used an open-loop structure to achieve the abovementioned recursive updating structure. The method we obtained is to input the pre-integration attitude value that has been updated synchronously with the gyroscope only from initialization in addition to the sensor data. The specific expression is as follows:(25)$$\begin{array}{cc} {\tilde{C}}_{b(k-1)}^{n} & =C_{b(0)}^{n}\cdot \exp([y_{Gyr,1}\times ]\Delta t)\cdot \exp([y_{Gyr,2}\times ]\Delta t)\cdot \ldots \cdot \exp([y_{Gyr,k-1}\times ]\Delta t)\\ & =C_{b(0)}^{n}\exp\left(\sum_{i=0}^{k-1}[y_{Gyr,i}\times ]\Delta t\right) \end{array}$$

The cumulative error of ${\tilde{C}}_{b(k-1)}^{n}$ can be merged into the error terms during the calculation of sensor compensation and eliminated together using the TCN denoising network through training. This enables the TCN to model gyroscope error alongside their own sensor error and serves as data augmentation to alleviate overfitting. In high-performance hardware systems, the above method can rely more on the direct attitude information obtained from the gyroscope, reduce the reliance on the magnetometer and accelerometer, and prevent erroneous correction.

We apply the same approach when calculating $e_{Mag}$, with the gravitational acceleration being replaced by accelerometer measurement $g^{\prime }\approx y_{Acc}$. This decouples the denoising neural network for the accelerometer and for the magnetometer, preventing the error propagation in multi-stage models.

The output of the ECF model is a corrected angular velocity ${\hat{\omega }}_{k}$. Additionally, the attitude update can achieve an open-loop state by using Equation (3). This is only necessary to obtain the initial attitude value $C_{b(0)}^{n}$, and the corresponding attitude value can be calculated in real-time in the subsequent integration process. The algorithm frame diagram is shown in Figure 2.

### 3.4. Loss Function

Since the proposed model needs to adapt to different application requirements of different hardware devices, designing a training and optimization method that meets most datasets is a necessary condition for the implementation of this method. We propose to use the form of incremental accumulation to integrate the attitude change to reduce the frequency of the IMU. The incremental change in attitude from time *i* to *i + j* can be calculated as:(26)$$\delta C_{b(i,i+j)}^{n}=C_{n(i)}^{b}\cdot C_{b(i+j)}^{n}=\prod_{k=i}^{i+j-1}\exp([\omega_{k}\times ]\Delta t)$$

We can merge *j* frames of attitude estimation data into one frame through Equation (26), and calculate the loss function with the label of the corresponding timestamps:(27)$${ℒ}_{j}=\sum_{i}\rho \left(\log\left(\delta C_{n(i,i+j)}^{b}\delta {\hat{C}}_{b(i,i+j)}^{n}\right)\right)\;$$
where the log(·) is the SO(3) logarithm map, and ρ is the smooth L1 loss, i.e., Huber loss. Since the calculation of the incremental change does not require the participation of the original attitude, and the left multiplication rule of SO(3) is followed during the transformation, the calculation of the loss function for the cumulative attitude change is not disturbed by the initial attitude. Finally, set the Huber loss parameter to 0.005, and j takes 16 and 32 to calculate the loss function:(28)$$ℒ={ℒ}_{16}+{ℒ}_{32}$$

## 4. Experimental and Results

### 4.1. Experiment Setup

In order to achieve accurate data-driven attitude estimation, it is necessary to have a relatively accurate 3D attitude gold standard to annotate the dataset. The calibrated VICON Optical motion capture system has been tested to achieve an accuracy of 0.1° [28], which is sufficient to calibrate our inertial equipment. To validate the designed attitude estimation model under different hardware devices and usage modes, this paper chose three public datasets for validation. Below, we will introduce their hardware systems and experimental methods.

#### 4.1.1. TUM VI Dataset

The TUM VI dataset [45] consists of visual–inertial sequences from different scenarios captured using hand-held devices. Different sensors and reflective markers are fixed on a rigid frame that can be handheld, and coordinate frame transformations are implemented to align the data from the IMU and markers in the same carrier right-hand b-frame, reducing the preprocessing difficulty. The inertial sensor used is a Bosch Sensortec consumer-grade six-axis IMU, model BMI160. It includes a three-axis accelerometer and gyroscope, with a sampling rate of 200 Hz and proper calibration and time synchronization. The ground truth is obtained using an optical motion capture system from OptiTrack, which contains 16 Flex13 cameras with a sampling frequency of 120 Hz. We select the room sequence set with the longest sequence as the training and test datasets. The sequences are entirely captured in the motion capture room, with the user holding the capture device and walking freely in the room, recording video images and IMU data. Each sequence lasts for 2–3 min. We define the training set as the first 80 s of sequences room1, room3, and room5; the validation set as the remaining end of these sequences; and the test set as the other three room sequences.

#### 4.1.2. EuRoC MAV Dataset

The EuRoC MAV dataset [46] consists of visual–inertial sequences deployed on Micro Aerial Vehicles (MAVs). Specifically, it includes synchronized stereo images, IMU data, and ground truth data provided by a Leica Nova MS50 laser tracker and Vicon motion capture system. The inertial sensor uses the ADI ADIS16448 6-axis IMU with a sampling rate of 200 Hz, without calibration, but with precise time synchronization between the sensor and ground truth data. The MAV collected 11 flight trajectories in 2 environments, a factory and a motion capture room, each lasting 2–3 min. Due to the flight characteristics of the UAV, it maintains dynamic motion throughout the recording process, and there is no obvious zero-velocity state available for traditional algorithms to converge, making attitude estimation challenging. We define the training set as the first 80 s of sequences MH (01, 03, 05), V1 (02), and V2 (01, 03); the validation set as the remaining parts of these sequences; and the test set as the remaining five sequences.

#### 4.1.3. OxIOD Dataset

The OxIOD dataset [47] consists of inertial motion sequences collected via smartphones. The inertial sensors use the ICM-20600 six-axis IMU from InvenSense and the HSCDTD004A three-axis magnetometer from Alps, and data are collected at a sampling rate of 100 Hz and 50 Hz for magnetometer, respectively. The ground truth system is collected using the Vicon motion capture system deployed indoors. This dataset provides a variety of phone wearing modes (handheld, pants pocket, handbag, and trolley), different types of motion (slow walking, normal walking, and running), various devices, and users. We chose all sequences containing different usage and motion types in the dataset and divided them according to the given training set/test set annotations from the Handheld (HH), Pocket (PO), Handbag (HB), Trolley (TR), Slowly Walking (SW), and Running (RU) datasets. We selected the first 80 s of the training set sequences as the training set and the remaining data as the validation set. In testing, due to the large variation in sequence length under different modes and the inevitable error accumulation caused by long-term operation, we tested all data for the first 2 min for comparison over different datasets.

We confirmed some issues with the dataset processing mentioned in [48]. Firstly, due to the fact that smartphones cannot access the raw sensor data, the motion data provided by the API is processed through the smartphone’s built-in attitude algorithm to eliminate gravitational acceleration or to convert the b-frame to the n-frame. The built-in attitude algorithm is generally composed of filtering algorithms, which perform badly under dynamic motion and magnetic disturbance. When using the ECF algorithm to process the OxIOD dataset, we discovered significant periodic disturbance, which affected the estimation of Roll and Pitch angles. We speculate that this is due to the magnetic disturbance present in the data collection room, which affects the API’s attitude calculation and provides unreliable IMU data. In addition, we analyzed that the smartphone and Vicon have different left- and right-hand b-frames, so we added a negative sign in front of the xy-axis of the gyroscope, the z-axis of the accelerometer, and the z-axis of the magnetometer to realize the coordinate frame transformation. In addition, we removed the HH_data5_seq4 test sequence because its motion data showed a state of significant disturbance, resulting in errors exceeding the normal range in the validation metrics.

### 4.2. Experiment Settings

We implemented the proposed network using the Pytorch framework and trained and deployed it on one NVIDIA RTX2080Ti. During training, we used the Adam optimizer with an initial learning rate of 0.01 and a cosine annealing learning rate adjustment strategy. To prevent overfitting, we set the weight decay parameter to 0.1 and the dropout parameter to 0.1. Additionally, we used Gaussian white noise to augment the data, which improved the model’s robustness and prevented overfitting. During training, the entire network had only 244,877 trainable parameters, and the model size was only 995 KB. We iterated over each dataset 1800 times and saved the model parameters that produced the smallest validation loss during training. During testing, since the network is entirely feedforward, it took only 0.07 s to process a 1-min (100 Hz sampling rate) input sequence, demonstrating real-time computational capabilities.

### 4.3. Metrics and Evaluation Protocol

In terms of evaluation methods, for test sequences of length *n*, we choose the following two evaluation criteria. The absolute attitude error (AAE) is calculated as follows:(29)$$\mathrm{AAE}=\sqrt{\sum_{i=1}^{n}\frac{1}{n}{\left\|\log\left(C_{n(i)}^{b}{\hat{C}}_{b(i)}^{n}\right)\right\|}_{2}^{2}}$$

Similar to the calculation of loss function, the attitude error is realized by multiplying the rotation matrix of the predicted attitude to the left by the transposition of the rotation matrix label, and then using the logarithmic mapping of SO(3) to the vector space, and then calculate the root mean square error (RMSE) (in °)on the sequence after taking the second-order norm of the error vector.
(30)$$\mathrm{AYE}=\sqrt{\sum_{i=1}^{n}\frac{1}{n}{\left\|\psi_{i}-{\hat{\psi }}_{i}\right\|}^{2}}$$

In addition to the AAE, the heading angle estimation error of the model is also an important performance indicator, which is vital in tasks such as PDR. The absolute yaw error (AYE) calculates the RMSE on the sequence after calculating the yaw error (in °) between the predicted heading angle and the ground truth.

When computing the sequence error, the estimated attitude is aligned with the ground truth attitude when *n* = 0, so that the result is only related to the model prediction. Due to the different lengths of test trajectories on different data sets, absolute error varies vigorously in different settings. In [49], a relative orientation error (ROE) evaluation method is proposed, which mainly divides the trajectory into small segments according to the travel distance, performs attitude alignment before the small segment, and then calculates the attitude error within the time interval. Since this experiment does not involve the prediction of the trajectory position, we use the AAE and AYE within the same time interval after the start for performance comparison during the test.

### 4.4. Performance on Attitude Estimation

First, we evaluate the attitude estimation performance on the TUM VI dataset. Since TUM VI only provides six-axis inertial motion data, we use the six-axis IMU version of the attitude estimation model, which only performs gyroscope noise reduction and external acceleration compensation. The results are shown in Table 1. We show the compared representative methods as follows:Raw IMU: Using raw uncalibrated gyroscope data and integrating directly after initial alignment to obtain attitude, this method is the benchmark for all attitude algorithms.ECF [25]: The state-of-art complementary filtering method introduced in Section 4.2, which is also our fusion framework.Calibrated IMU: A total of 45 calibration parameters and fusion gains of the gyroscope, accelerometer, and magnetometer are automatically generated through a data-driven method, and the corresponding parameters are all non-zero constants. This method can be seen as setting the sensor input data of the neural network to 0 during training and only training the calibration parameters.OriNet [33]: A 3D pose estimation method based on a LSTM network.DIG [34]: The gyroscope noise reduction network proposed by Brossard et al. integrates the angular velocity data after noise reduction to obtain the attitude value and does not perform extra processing on large external acceleration and magnetic disturbance.

As shown in Table 1, the Raw IMU achieved a relatively small attitude and heading angle error, which is due to the high sampling rate and preliminary calibration of the IMU used in TUM VI. The static error of the gyroscope is within a small range, and the data collection process is mainly based on human walking, which is relatively smooth and does not amplify the corresponding dynamic error. After fusing the acceleration data, the ECF’s performance decreased compared to the baseline, because the uncalibrated accelerometer introduced new sources of error. In the case where the gyroscope data are more accurate and there is no dynamic change in the fusion gain *K*, overcorrection occurred. The error was significantly reduced using the data-driven calibration method, and the Calibrated IMU yielded better performance than OriNet and DIG due to the low dynamic error of the usage scenario. The proposed method absorbs the advantages of the above methods, and after static and dynamic denoising of the two sensors, fuses them through the ECF framework to achieve corrected angular velocity data, with the framework having been experimentally demonstrated to perform well in attitude estimation tasks. Compared to baseline method DIG, the proposed method achieved AAE and AYE reduction for 41% and 45% in the TUM VI dataset. Figure 3 and Figure 4, respectively, show the attitude angle changes and attitude angle errors for the test sequence room6. It can be seen that the Raw IMU and ECF have large fluctuations in the Roll and Pitch angles, and the error of the yaw angle increases over time due to the lack of heading calibration information. However, our method based on denoising neural networks can achieve higher estimation accuracy by autonomously compensating for the yaw angle.

### 4.5. Performance under Dynamic Motion and Magnetic Disturbance

In this section, we assess the effectiveness of the proposed 6-axis attitude estimation in the face of persistent large external accelerations using the EuRoC MAV dataset. We perform a comprehensive evaluation of our nine-axis attitude estimation model on the OxIOD dataset, incorporating noise reduction for gyroscope, compensation for external accelerations, and compensation for magnetic disturbance. The results are shown in Table 2.

As shown in Table 2, in the comparison of the two difficult motion sequences MH_02 and V1_03, the proposed method has achieved high performance, but a relatively large error occurred in the simple sequences, such as with MH_02 and V1_01. The error is mainly composed of the yaw angle error. Our analysis is that the movement trajectory in the EuRoC MAV dataset is relatively single, and the six-axis attitude estimation model has no calibration information at the heading, which leads to error accumulation. Overall, our method improves the baseline method DIG under dynamic motion by 23.4% and 23.9% for AAE and AYE. Since the EuRoC MAV dataset did not calibrate the static bias of the gyroscope beforehand and had no long stationary phase for attitude error correction, the ECF method resulted in significant errors in both the AAE and AYE metrics compared to data-driven methods.

Figure 5 shows the change in the attitude angle in the V1_03 sequence. It can be seen that due to the calibration information of the accelerometer, the error of the ECF attitude estimation is reduced compared to the Raw IMU method, but there is still a certain distance from the true value. The noise-reduction neural network suffers from severe movements, such as rapid turning, when the accuracy decreases, which causes the gap between the yaw angle curve in the figure and the true value at the peak.

For the OxIOD dataset, the proposed method achieved the best performance in terms of average AAE and AYE metrics for different sequences, achieving performance improvement by 36.3% and 37.8%. However, it can be observed that the performance of the network is poorer in the Running mode. The LSTM-based denoising method also achieved good results on this trajectory. We analyzed that this is because the Running mode involves a change in the motion pattern, and the LSTM network, which is designed to handle time series relationships, has stronger time feature extraction capabilities compared to the 1D-CNN network. Additionally, the forget gate design allows for longer time series to have better memory. All methods exhibited significant attitude errors in the HH_data5_seq3 sequence. Through analysis of the curves of the three attitude angles over time, it was determined that there exists a stable constant error between the predicted yaw angle and the true value, indicating that the original sensor data were interfered with during the recording of this sequence. For sequences with a smoother motion pattern, such as SW_d1_seq8 and TR_d2_seq6, the Raw IMU method showed better performance, while all the other data-driven methods seem to bring extra errors to the attitude estimation. The data-driven methods appear to overfit difficult cases; however, they cause accuracy degradation in easy cases. Nevertheless, our proposed method outperforms the ECF algorithm by 34.0% and 34.5% in two metrics. We add the attitude calculated by the internal API of the mobile phone recorded in the dataset for comparison (purple curve). The attitude estimation curves of various methods on the PO_data2_seq6 trajectory are shown in Figure 6.

As shown in Figure 6, the attitude data obtained using the API exhibit significant drift in the Roll and Pitch angles. We analyzed that this was due to the circular indoor trajectory, where there were stable sources of magnetic disturbance. Each time the trajectory approached the disturbance source, it affected the magnetometer data, thus impacting the internal attitude estimation algorithm. Specifically, this manifested as a periodic disturbance of approximately 20 s added to the stable periodic walking where the actual frequency was about 0.73 Hz. Similarly, when using the Raw IMU and ECF to calculate the inertial motion data affected by unreliable attitude estimation algorithms, corresponding magnetic disturbance would also occur. Accordingly, the proposed denoising attitude estimation model achieved excellent magnetic disturbance elimination, thereby improving the robustness of attitude estimation.

### 4.6. Generalization

In order to evaluate the robustness of our model on unseen sequences, considering the scenarios of commercial applications, we have adopted a set of tests using sequences from different users and devices provided by the OxIOD dataset. We use the trained model from Section 4.4, which used the dataset collected by User1 using an iPhone 7 Plus smartphone, with the test set comprised of four different users and two different smartphone devices. The results of AAE and AYE of the test sequences are visualized in Figure 7. We have also provided the results of trained nine-axis input version of DIG and ECF which parameters are optimized in training sequences for comparison.

As illustrated in Figure 7, the proposed AHRS generally outperforms the baseline methods. However, in some test sequences such as User3, the data-driven methods appear to have a large variance caused by a domain shift. The average AAE and AYE of the proposed AHRS on 52 test sequences are 20.3° and 11.2°, which are lower than both benchmarks, showing the capability of generalizability over different users and devices. However, the generalization ability of the proposed network to different types of IMU is still challenging, and we would like to try more datasets and data augmentation techniques to improve the generalization of the proposed model.

## 5. Discussion

### 5.1. The Effectiveness of Data-Driven Error Compensation Model

After comparing the results of different methods in Table 1 and Table 2, we can conclude that modeling static errors, such as in the Calibrated IMU method, can significantly improve attitude estimation performance. Furthermore, modeling dynamic errors with neural network models can further enhance the performance and robustness of the model. In addition, we can compare the difference between the denoised angular velocity output of the model and the raw angular velocity output of the gyroscope. Figure 8a,b show the curves of this difference in the TUM VI dataset room6 sequence and the EuRoC MAV dataset V1_03 sequence. We can see that the two curves have different y-axis intercepts, which is because the TUM VI dataset calibrates the IMU and eliminates static bias errors, while the EuRoC MAV dataset does not. As we did not set a separate parameter for calibrating the static bias of the gyroscope, the calibration of the static bias shown in Figure 8b is entirely learned through the neural network.

The DIG method only models and denoises the error data of the gyroscope and cannot completely eliminate errors in the presence of significant magnetic disturbance and large external acceleration. In this section, we conducted ablation experiments on the error calibration model of the Calibrated IMU method on the OxIOD dataset, specifically comparing the performance difference between calibrating all sensor parameters and only calibrating gyroscope parameters. The comparison results are shown in Figure 9. It is evident that calibrating all parameters leads to a significant improvement in performance.

### 5.2. Different Input of Gyroscope Denoising Neural Network

Equations (16)–(19) discussed the principle that accelerometers and magnetometers affect the estimation of angular velocity through attitude computation, and concluded that accelerometer and gyroscope data should be included as input to the gyroscope noise prediction network. Here, we analyze the effectiveness of this theory by comparing the performance of the gyroscope denoising model when nine-dimensional data (including accelerometer and magnetometer data) are input versus only three-dimensional gyroscope data. Other network parameters were kept the same. The comparison results confirmed the effectiveness of this method and are shown in Figure 10.

### 5.3. Ablation Experiment of Different Compensation Source

We conducted ablation experiments on different components of the model, including the external accelerations compensation and magnetic disturbance compensation models, on the OxIOD dataset to analyze the effectiveness of each component. The comparison results are shown in Table 3. It can be seen that the proposed model components showed significant performance improvements compared to the baseline model on most sequences, and the model can adaptively adjust its dependence on different error eliminations to achieve improved accuracy after fusion. The results of the ablation experiments suggest that the model applies different components to handle different motion errors, and the joint action of the two error elimination methods improves the overall accuracy of attitude estimation.

## 6. Conclusions

In this study, we present a robust AHRS algorithm for accurate attitude estimation in dynamic motion and magnetic disturbance scenarios. We propose a data-driven IMU calibration model to compensate for systematic and random errors in the raw sensor data, providing denoised angular velocity, gravitational acceleration, and a geomagnetic north field. To estimate random error terms such as gyroscope random bias, external acceleration, and magnetic disturbance, we use TCNs as feature extractors with raw IMU readouts as input. By optimizing the systematic and random errors of the IMU using backpropagation during training, our model can accurately estimate attitude. The ECF framework is used for sensor fusion, and we modify the close-loop iteration, turning it into an open-loop estimation, to accelerate the training process. We decouple different error compensation networks to prevent the accumulation of error and erroneous correction in different usage scenarios. We conduct extensive experiments to evaluate our proposed model on motion data under dynamic motion and magnetic disturbance, achieving more than 23.4% and 23.9% performance improvement compared to the baseline data-driven method and 34.0% and 34.5% compared to the ECF traditional filter. Generalization experiments on different devices and users demonstrate the robustness of our model to adapt to different scenarios.

Future work includes adapting our model to more sophisticated datasets covering daily usage scenarios with more motion modes, devices with different sensors and sampling frequencies, quantitative environmental condition changes, and outdoor conditions. We aim to improve the generalizability of our model over different datasets under the same network parameters using transfer learning and other data augmentation techniques. To deploy our model on low-end smartphones and embedded systems, further optimization and compression techniques are needed, such as knowledge distillation and quantization.

## Figures and Tables

**Figure 1 micromachines-14-01070-f001:**
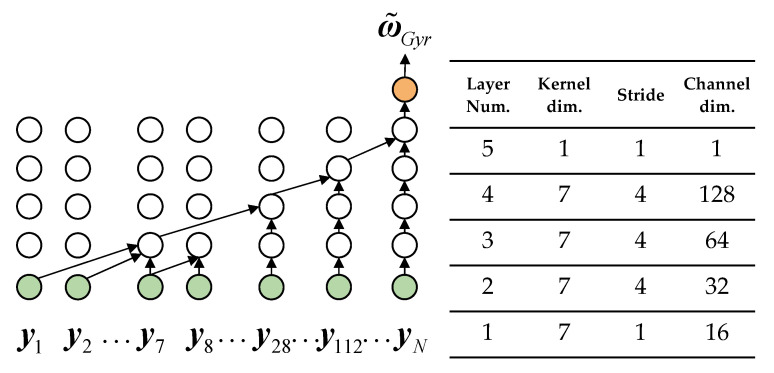
The structure and detailed parameters of $f_{TCN}$. The 5-layer structure has a receptive field of N = 448, which can be calculated by the product of the 1D CNN kernel size and multiplied by each layer’s stride.

**Figure 2 micromachines-14-01070-f002:**
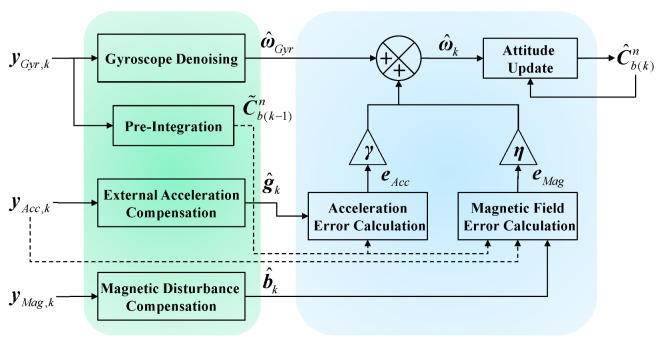
Complete block diagram of the proposed AHRS. The green part represents the data-driven IMU calibration model, and the blue part represent the open-loop ECF estimator.

**Figure 3 micromachines-14-01070-f003:**
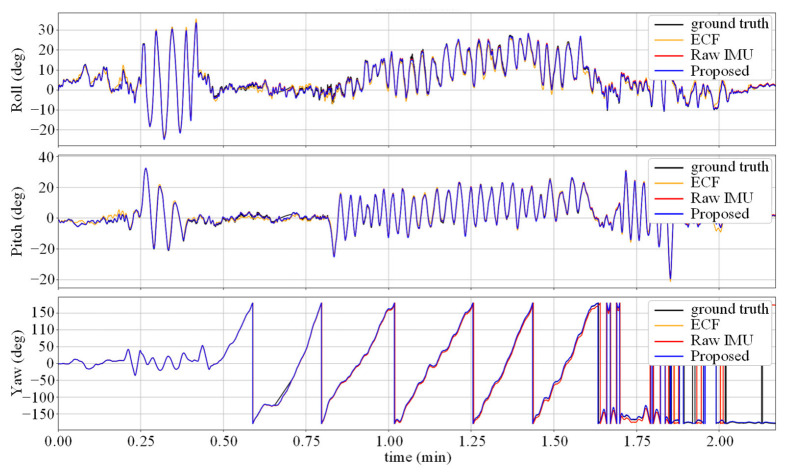
Attitude estimates for the test sequence room6.

**Figure 4 micromachines-14-01070-f004:**
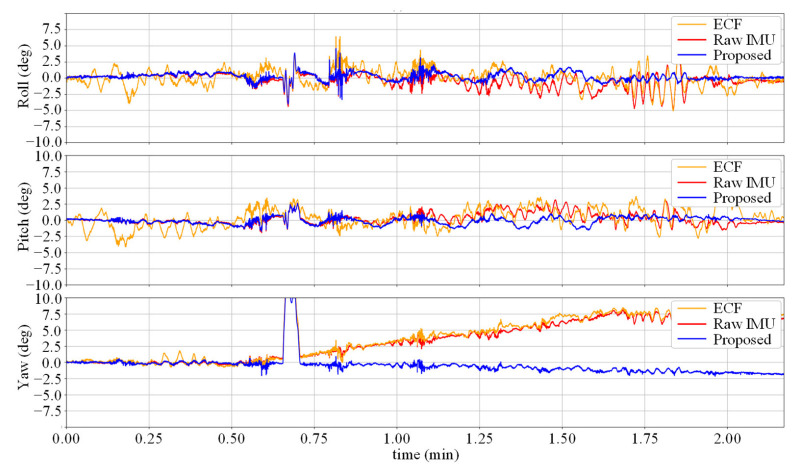
Attitude errors for the sequence room6. Our method removes long term accumulative error compared to the ECF algorithm.

**Figure 5 micromachines-14-01070-f005:**
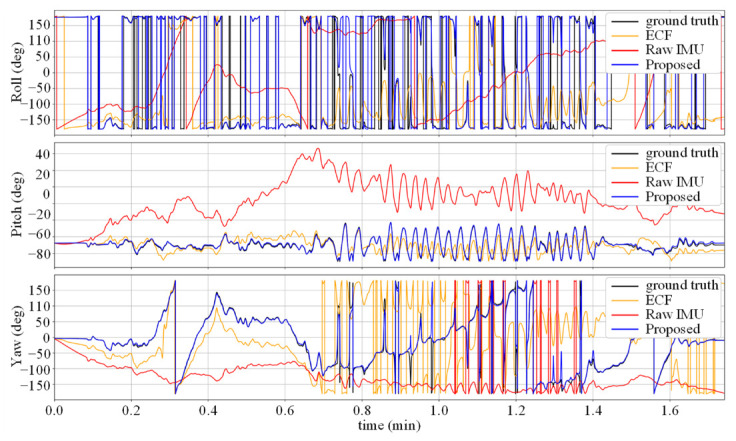
Attitude estimates for the difficult test sequence V1 03.

**Figure 6 micromachines-14-01070-f006:**
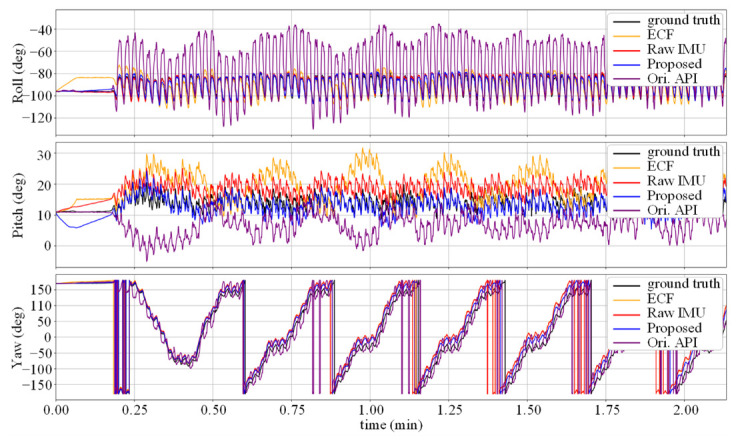
Attitude estimates on the test sequence PO_data2_seq6. The API’s attitude exhibits periodic errors and drifts compared to the ground truth.

**Figure 7 micromachines-14-01070-f007:**
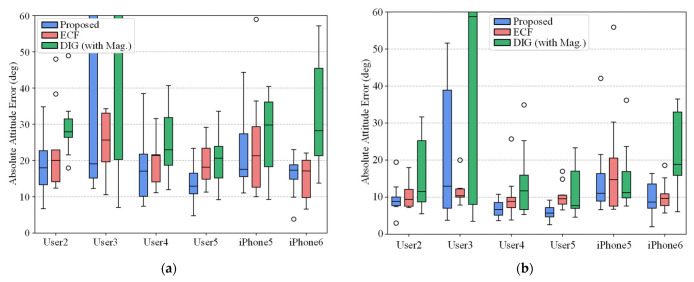
Generalization experiment on unseen sequences collected by different users and smartphone devices in OxIOD datasets. The errors are expressed in (**a**) AAE and (**b**) AYE.

**Figure 8 micromachines-14-01070-f008:**
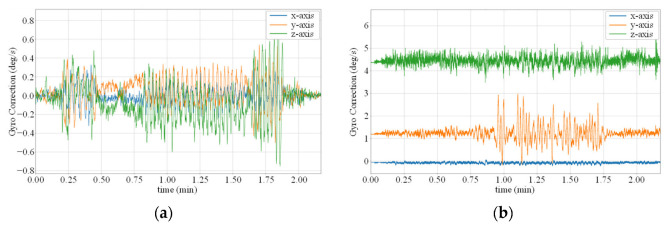
Gyroscope noise error term ${\tilde{\omega }}_{k}$ in two different datasets. (**a**) TUM VI dataset room6 test sequence; (**b**) EuRoC MAV dataset V1_03 test sequence.

**Figure 9 micromachines-14-01070-f009:**
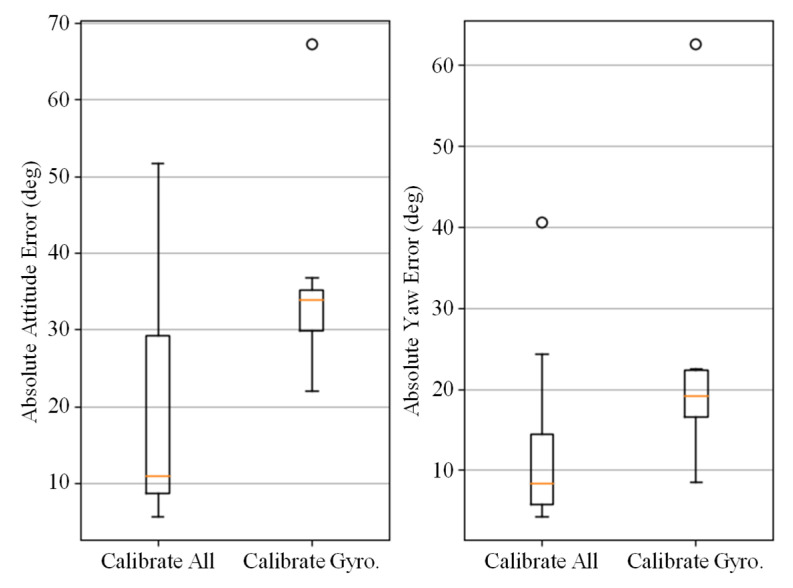
Comparison of Calibrate IMU method with all error terms calibrated and only the gyroscope errors calibrated. The errors are expressed in AAE (**left**) and AYE (**right**).

**Figure 10 micromachines-14-01070-f010:**
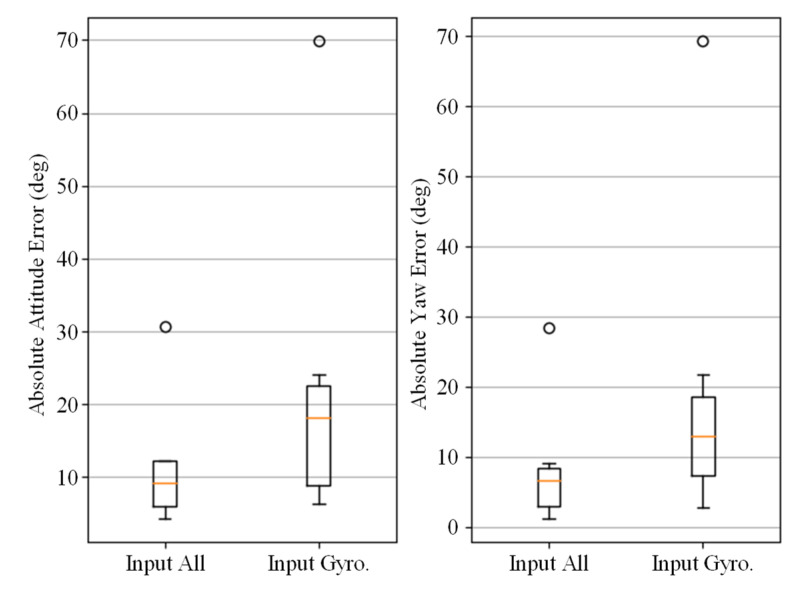
Comparison of gyroscope denoising neural network with all 9-axis IMU data input and only gyroscope data input. The errors are expressed in AAE (**left**) and AYE (**right**).

**Table 1 micromachines-14-01070-t001:** Absolute attitude error (AAE) and absolute yaw error (AYE) in °, on the test sequences of the TUM VI dataset.

Sequence	Raw IMU		ECF		Calib-IMU		OriNet		DIG		Proposed	
	AAE	AYE	AAE	AYE	AAE	AYE	AAE	AYE	AAE	AYE	AAE	AYE
room2	11.27	10.85	14.58	13.83	2.39	2.21	3.03	2.86	4.06	3.85	**2.06**	**1.47**
room4	2.51	2.13	3.86	2.80	**0.76**	**0.42**	0.81	0.49	1.12	0.73	0.88	0.59
room6	5.00	4.78	5.51	5.12	2.22	2.01	2.53	2.31	2.56	2.34	**1.95**	**1.74**
**average**	6.26	5.92	7.98	7.25	1.79	1.55	2.12	1.89	2.58	2.31	**1.63**	**1.27**

**Table 2 micromachines-14-01070-t002:** AAE and AYE results for the test sequences of the EuRoC MAV and OxIOD datasets.

Dataset	Sequence	Raw IMU		ECF		Calib-IMU		OriNet		DIG		Proposed	
		AAE	AYE	AAE	AYE	AAE	AYE	AAE	AYE	AAE	AYE	AAE	AYE
EuRoC MAV	MH 02 easy	123	104	80.19	18.87	5.21	3.89	6.48	3.12	7.86	4.92	**4.06**	**2.81**
	MH 04 difficult	130	108	62.98	15.83	4.26	3.57	4.64	3.41	**3.11**	**1.31**	3.31	1.39
	V2 02 medium	116	85	58.26	16.29	4.67	3.10	3.27	2.26	3.74	2.62	**2.89**	**2.19**
	V1 03 difficult	119	77	57.92	17.37	2.50	1.72	3.36	1.85	2.58	0.92	**2.30**	**0.80**
	V1 01 easy	114	80	59.38	12.6	5.73	4.58	6.13	4.47	5.13	2.99	**4.59**	**2.50**
	**average**	120.4	90.8	63.75	16.19	4.47	3.37	4.78	3.02	4.48	2.55	**3.43**	**1.94**
OxIOD	HB_d2_seq4	12.88	6.90	9.07	6.91	51.72	24.27	9.06	8.25	30.40	12.28	**8.43**	**5.71**
	HH_d5_seq1	**5.99**	**5.02**	6.08	5.49	9.52	7.17	7.30	5.69	12.86	10.21	9.37	5.26
	HH_d5_seq2	95.62	95.67	11.5	9.09	5.72	4.54	9.15	8.06	**3.77**	**3.17**	6.56	4.75
	HH_d5_seq3	100.13	100.36	57.84	56.23	42.94	40.62	51.45	49.62	37.90	35.21	**30.68**	**28.50**
	PO_d2_seq6	99.42	90.60	28.76	8.37	24.70	6.15	27.54	6.83	10.66	3.74	**5.02**	**2.34**
	RU_d1_seq7	8.06	**2.94**	7.37	2.95	6.30	4.29	**5.28**	3.87	9.65	7.90	12.21	9.07
	SW_d1_seq8	2.57	1.26	**2.21**	**0.81**	10.29	9.70	4.53	4.32	4.88	4.48	4.30	1.19
	TR_d2_seq6	2.95	1.72	2.64	1.72	11.70	11.23	**2.34**	**1.50**	19.77	19.41	6.23	3.20
	**average**	40.95	38.06	15.68	11.45	20.36	13.50	14.58	11.02	16.24	12.05	**10.35**	**7.50**

**Table 3 micromachines-14-01070-t003:** Ablation experiment results.

Sequence	Proposed		w/o Mag.		w/o Acc.		w/o Mag. and Acc.	
	AAE	AYE	AAE	AYE	AAE	AYE	AAE	AYE
HB_d2_seq4	**8.43**	**5.71**	36.31	18.47	23.9	11.51	35.15	12.04
HH_d5_seq1	**9.37**	**5.26**	9.02	5.84	11.18	6.71	7.5	4.97
HH_d5_seq2	**6.56**	4.75	6.77	**4.49**	7.98	5.31	10.24	8.62
HH_d5_seq3	**30.68**	**28.50**	42.49	38.66	54.82	52.66	55.34	53.51
PO_d2_seq6	**5.02**	**2.34**	16.78	7.85	16.48	5.93	19.88	5.6
RU_d1_seq7	12.21	9.07	**10.05**	**6.47**	11.54	8.21	33.65	30
SW_d1_seq8	4.30	1.19	3.68	1.64	**1.78**	**0.97**	13.59	12.91
TR_d2_seq6	**6.23**	**3.20**	8.71	6.58	12.89	11.3	13.95	13.72
**average**	**10.35**	**7.50**	16.73	11.25	17.57	12.83	23.66	17.67

## Data Availability

The experiments are performed on publicly available datasets. The sources for utilized datasets are available in [45,46,47].

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
