# Peer review of "Robust Attitude and Heading Estimation under Dynamic Motion and Magnetic Disturbance"

_micromachines, 2023, doi:10.3390/mi14051070_

Round 1

Reviewer 1 Report

The authors present a combination of existing methods for absolute orientation estimation. They employ an extended complementary filter from Madgwick et al. (Ref. 20) in an open-loop and error-decoupled manner, and apply a data-driven denoising for gyroscope, accelerometer and magnetometer measurements with a similar approach as presented in Brossard et al. (Ref. 26). The method is evaluated on multiple freely available datasets and evaluates the method accuracy with respect to a selected set of orientation estimation algorithms. The method is shows to achieve only comparable attitude and heading estimation, in comparison with the presented baseline methods. In summary, it seems like extensive processing methods are employed to gain little or no accuracy in the orientation estimation task.

Major comments:

-       Multiple recent and advanced methods for IMU-based orientation estimation were not addressed in previous work nor compared against. Only to mention a few and openly available recent and advanced methods: (Kok et al. 2019), (Laidig et al. 2022, (Weber et al. 2021).

-       A neural-network based approach was used for denoising raw sensor measurements. In order to proof robustness to overfitting, the method should be proven to generalize, for example to unseen datasets, that were not used during training. Multiple existing datasets, that were not considered can be used for this purpose, e.g., BROAD (Laidig et al., 2021.) and I suggest the authors apply this commonly used evaluation technique.

-       It is unclear how the denoising approach exactly achieves the robustness against the following, and how this exactly complements the employed approaches from Madgwick et al. and Brossard et al.:

o   effectively compensating for instantaneous accelerations and decelerations on the raw measurements of the specific force from the accelerometer.

o   Compensating for magnetic disturbances on the magnetometer readings.

-       It is unclear how the presented open-loop and error-decoupled version of the method from Madgwick et al. helps in overcome the problems addressed.

-       The authors point out in the abstract (Ln 21) that the proposed AHRS method ‘significantly’ improves accuracy and robustness under different scenarios. There are no statistical test provided that justify this wording. The statement does furthermore not coincides with the presented results, which are comparable to the utilized baseline methods in the presented work.

-       [Section 2. Extended Complementary Filter-based AHRS.] This section is only a repetition of existing work. I suggest to omit this part completely, if no additional contributions are made to the work.

-       Figure text / font size is way to small the plots and figures are generally of poor unreadable and unacceptable resolution.

I furthermore provide a list of additional minor comments that can be used to further improve the manuscript.

Minor comments:

- English wording and formulation of sentences should be improved in the manuscript.
- Sentences should be checked on exact intended meaning that the authors try to convey.

-  Many abbreviations are introduced. Please remove abbreviations if they are afterwards not or rarely used again in the manuscript.

Ln 11. The application range that is identified is rather small (“ pedestrian dead reckoning (PDR) and Micro Aerial Vehicles”). Inertial orientation estimation is the most crucial element that determines the accuracy of further derived metrics. Many applications benefit from this, e.g., motion tracking of human segments, robotic manipulators, autonomous driving, wearable robotics, … just to name a few more dimensions. I'd suggest to use the wording 'for example' / or ‘most notably’ if you only address the most common application domains or a shorter list.

Ln 46. ‘Gravity vector measured by accelerometer’ A MEMS accelerometer typically measures the specific force that is combination of a dominating gravity component with potentially superimposed linear accelerations and decelerations.  

Ln 56. ‘These interfering factors vary with the region of use and can affect the heading prediction of AHRS, thereby reducing positioning accuracy in PDR and other applications.’ The highlighted sentences breaks the reading very abrubtly.

Ln 61. ‘Early sensor fusion methods’ I’d suggest to use the wording ‘traditional’ or even ‘conventional’, since most currently methods do make use of a Kalman / Complementary approach.

Ln 67. ‘The classic Mahony app..’ I’d suggest the wording ‘Often adopted’ for example. The word classic may refer to the reason for its wide adoption, namely the free availability of the method online.

Ln 68. ‘Representative algorithm’ This wording relates to my previous comment and holds a subjective note that does not add value to the work presented.  

Ln 84. “When the algorithm detects dynamic motion or magnetic disturbance, it automatically disables the correction information … ” This feels fundamentally incorrect or would only work for very short periods of time of a couple <10 seconds in MEMS based gyroscope, before drift will be notably visible.

Ln 110 – 114 describe the contribution in brief which does not completely matches the section 1.2 which is used to describe the contribution in more detail. This is rather confusing for the reader.

Ln 575 the wording ‘accurate attitude estimation’ is not quantified with numbers. It would be great if a number for improvement in both the attitude and heading part of the orientation was provided. Potentially even for summarized groups of motion types / disturbances.

References:

Kok, Manon, and Thomas B. Schön. "A fast and robust algorithm for orientation estimation using inertial sensors." IEEE Signal Processing Letters 26.11 (2019): 1673-1677.

Laidig, Daniel, et al. "VQF: A Milestone in Accuracy and Versatility of 6D and 9D Inertial Orientation Estimation." 2022 25th International Conference on Information Fusion (FUSION). IEEE, 2022.

Weber, Daniel, Clemens Gühmann, and Thomas Seel. "RIANN—A robust neural network outperforms attitude estimation filters." Ai 2.3 (2021): 444-463..

Laidig, Daniel, et al. "BROAD—A benchmark for robust inertial orientation estimation." Data 6.7 (2021): 72.

Author Response

Dear Reviewer,

Thank you very much for your valuable and constructive comments on our manuscript. We totally agree with you that major revisions need to make to our manuscript. We have tried our best to improve the manuscript according to your comments. 

Please find the attached file for our point-to-point response.  

Best Regards 

Reviewer 2 Report

Dear Authors,

Please find the attached file for your reference. Please update the paper based on the comments and resubmit it. 

Regards 

Author Response

Dear Reviewer,

Thank you very much for your valuable and constructive comments on our manuscript.  We have tried our best to improve the manuscript according to your comments.

Please find the attached file for our point-to-point response.  

Best Regards 

Round 2

Reviewer 2 Report

Dear Authors,

Thank you for addressing all my comments and I don't have any further concerns on your paper.

Regards